# Evidence based targeting of districts for active surveillance of skin-related neglected tropical diseases in Ghana

William Jones-Warner[1]*, Yaw Ampem Amoako[2,3], Joseph Opare[4],
Nana Konama Kotey[4], Dorothy Yeboah-Manu[5], Richard Odame Phillips[2],
Rachel Pullan[1], Hope Simpson[1,6]

1 Department of Disease Control, London School of Hygiene and Tropical Medicine, London, United Kingdom, 2 Kumasi Centre for Collaborative Research in Tropical Medicine, Kwame Nkrumah University of Science and Technology, Kumasi, Ghana, 3 School of Medical Sciences, Kwame Nkrumah University of Science and Technology, Kumasi, Ghana, 4 Ghana Health Service, Neglected Tropical Diseases Programme, Accra, Ghana, 5 Department of Bacteriology, Noguchi Memorial Institute for Medical Research, University of Ghana, Accra, Ghana, 6 Department of Global Health and Infection, Brighton and Sussex Medical School, Falmer, United Kingdom

* William.jones-warner@lshtm.ac.uk

## Abstract

To improve control and management of skin-neglected tropical diseases (NTDs), district-level integration of case finding and management is recommended. However, these strategies are costly and should be targeted to co-endemic areas. Identifying districts with high burdens of undiagnosed cases, particularly where access to healthcare is limited, can better direct efforts. We developed an evidence consensus framework—a structured decision-making approach that combines information from multiple sources to support decision-makers. Using this approach, we built an interactive dashboard that brings together data on factors such as indicators of disease endemicity from model predictions, population vulnerability to disease, access to and availability of health services, and risk factors for poor clinical outcomes. Each factor is given a score, which is then adjusted so that no single type of information outweighs the others. Districts were scored and ranked based on levels of each indicator, and districts scoring highest across combined criteria were identified. We visualised results on an interactive dashboard, or webpage, intended for use by decision-makers in NTD programs in Ghana. We identified 108 districts potentially endemic for both Buruli ulcer (BU) and lymphatic filariasis (LF). Of these, 17 districts ranked in the highest quintile for overall score and were deemed suitable for active case detection of skin NTDs. Notably, Pru East, Shama, and Nzema East scored highest, despite mixed BU endemicity. Six districts, including Shama, Awutu Senya East, and Ekumfi, scored high for both BU and LF, making them priority areas for active BU detection. This evidence-based framework offers a practical method for integrating datasets to guide surveillance and decision-making in skin NTDs. It emphasizes prioritizing districts with high overall scores and predicted LF or BU

License, which permits unrestricted use, distribution, and reproduction in any medium, provided the original author and source are credited.

**Data availability statement:** All relevant data for this study are publicly available from the GitHub repository (https://github.com/Williamjoneswarner/Skin-NTD).

**Funding:** This study was funded by American Leprosy Missions (AIM Initiative). The funders had no role in study design, data collection and analysis, decision to publish, or preparation of the manuscript.

**Competing interests:** The authors have declared that no competing interests exist.

prevalence, while addressing gaps in knowledge about BU risk factors. By simplifying data integration, this framework enhances surveillance efforts, improving coverage and resource allocation.

---

## Author summary

Skin-related neglected tropical diseases (NTDs), such as Buruli ulcer and lymphatic filariasis, are often under-reported because they affect remote communities with limited access to healthcare. Finding these cases early is important, but "active case detection" can be costly. To make the best use of limited resources, we developed a practical approach to identify the districts in Ghana where case-finding is most likely to have an impact. We began by building a **conceptual framework**—a structured flowchart to think about where undiagnosed cases might be found—made up of five key domains: 1. **Evidence for endemicity**, 2. **Population vulnerability to disease**, 3. **Accessibility of health services**, 4. **Availability of health services**, 5. **Risk factors for poor clinical outcomes.** For each domain, we identified contributing factors (e.g., predicted disease risk, poverty levels, travel time to clinics, number of health centres, and under-five mortality) and found open-source datasets that could represent them. We summarised these data for each district, converted them into comparable scores, and gave each domain equal weight to ensure a balanced final ranking. We then presented the results in an **interactive online dashboard (website)** so that decision-makers can explore district-level maps and scores without needing specialist mapping software. Using this method, we identified 17 districts in Ghana as top priorities for active case detection of skin NTDs. Because the approach uses only open-source data, it can be adapted for other countries, different geographic scales, or even other diseases. This makes it a flexible tool for public health teams to target interventions where they are most needed.

## Introduction

Neglected tropical diseases affecting the skin (referred to as skin NTDs) constitute a subset of ten conditions outlined by the World Health Organization (WHO). These diseases manifest through various alterations in the skin [1]. They are part of a diverse group of illnesses prevalent mainly in low- and middle-income countries (LMICs) and underserved communities. Historically, these diseases have received limited attention from public health initiatives, research actions, and drug development efforts, hence the term "neglected". Collectively, they exert a significant detrimental impact on the health and overall well-being of affected populations.

By leveraging the shared characteristics of impacting the skin, and demographic risk factors, the integration of skin NTD management has emerged as a pivotal focus area identified to enhance the delivery of effective outcomes outlined in the WHO

roadmap. This strategic shift departs from traditional disease-specific approaches that sometimes resulted in duplicated efforts and competing demands for scarce resources. The framework was devised with the objective of identifying opportunities for integration across various skin NTDs. Its purpose is to enable endemic countries to adapt their current recommendations to seamlessly incorporate them into their national health plans and programs [2,3].

The skin NTDs include Buruli ulcer (BU); cutaneous leishmaniasis (CL); mycetoma, chromoblastomycosis and other deep mycoses; leprosy (Hansen's disease); lymphatic filariasis (LF); onchocerciasis; post-kala-azar dermal leishmaniasis; scabies and other ectoparasitoses (including tungiasis); onchocerciasis [4] and yaws [1]. While many NTDs can be treated by the use of mass drug administration (MDA) [5], skin NTDs generally require individual diagnosis and case management, often necessitating prolonged periods of care [3,6]. If left untreated, they can lead to lifelong disfigurement, disability, socioeconomic difficulties and negative impacts on mental health [7] further limiting the capacity of individuals in lower socioeconomic circumstances to thrive.

Data on the full burden and distribution of skin NTDs are limited, as these diseases are underdiagnosed and underreported for various reasons [8]. Firstly, they often affect communities where healthcare access is limited [9]. Even if health facilities are accessible, they may lack staff trained in skin NTD diagnosis and management, diagnostic tools, medicines and medical supplies, or tools and procedures for recording and reporting, any of which could result in cases not being notified. Their distributions vary in extent and focality but often overlap due to shared poverty-related risk factors, which mean that resource deprived, marginalized and remote communities are disproportionately affected [2]. Furthermore, people affected may decide not to seek formal medical attention due to a lack of awareness, cultural beliefs, stigma, or the absence of healthcare infrastructure [9–12]. Initially, skin NTDs can exhibit mild and non-specific symptoms, causing those affected to overlook them or turn to traditional remedies rather than seeking medical help. Consequently, the prevalence of skin NTDs is underestimated in passive surveillance; for instance, one study found that only one in thirteen active BU cases was reported at the hospital. This indicates that individuals often choose their own methods for self-reporting symptoms and seeking healthcare [12,13].

Therefore, the WHO has recommended that active case detection should be conducted to promote early detection and treatment of skin NTDs and improve understanding of their burden and distribution [10,13–15]. Implementation of active case detection for NTDs poses significant challenges, including high costs and low resource availability, necessitating careful targeting of efforts [16]. One method to mitigate the financial burden associated with comprehensive screening campaigns, diagnostic tools, and treatment cost is to integrate activities across multiple diseases, for example at the district level [6,15]. District-level integrated management strategies for skin NTDs are currently being implemented and evaluated in Ghana and Ethiopia [3,17].

The limited availability of accurate and up-to-date information on disease prevalence and distribution makes it challenging to identify high-risk areas and deploy case-finding activities and interventions strategically in the areas that need them most urgently. Available epidemiological data include cases reported in the published literature, and those detected through national surveillance programs, although the latter are not usually made publicly available at subnational levels [18]. Alternative sources of information may support identification of potentially endemic areas represented by epidemiological data. These information sources include disease distribution models and open-access datasets derived from satellite imagery. In recent years, environmental and geostatistical models have been used to predict distribution or prevalence of skin NTDs using environmental covariates associated with recorded cases or survey prevalence. Examples at continental and global levels include the predicted distribution of Buruli ulcer [19] and podoconiosis [20] and the predicted prevalence of lymphatic filariasis [21–23] and onchocerciasis [24]. There are also examples of single-country models for various skin NTDs [19,21,25,26]. Utilisation of predicted disease prevalence or endemicity to target active case detection would allow for resources to be focused on areas with a higher likelihood of disease occurrence, without restricting interest to areas where cases have already been detected and reported, which are likely to be those with better coverage of treatment and control programs.

For other skin NTDs which do not have such strong environmental drivers (such as yaws, scabies, and leprosy), predictive models have not been developed and it is unlikely that they would be reliable. However, spatial datasets may still offer useful information to indicate areas which may be at higher risk. For example, yaws may be more likely to occur in rural areas [27], and skin NTDs in general are expected to be concentrated in economically deprived communities. Spatial datasets may also identify locations where cases are less likely to have been detected- for example, places with low accessibility to health facilities.

Spatial targeting of interventions to predicted malaria hotspots has been discussed as a potential method for reducing transmission [28], and although evidence for an effect on transmission is limited [29], it may still play a useful role in improving case detection. Uptake of model predictions to inform case finding approaches for skin NTDs has been limited, and there are probably multiple reasons for this. One reason may be the uncertainty associated with these models, as case searches targeted on predicted distributions may not identify any cases if locations are wrongly predicted to be endemic (or were once but are no longer endemic). Another reason may be that the formats in which predictions are presented are difficult to manage and interpret.. Distribution predictions are usually presented as continuous gridded datasets—often in the form of a raster file, which is a map made up of evenly spaced squares, each containing a value such as predicted prevalence. To extract useful information at the district or regional level, these files need to be opened in specialist mapping software or geographic information system, such as QGIS or ArcGIS.

In this report we explore the use of multiple open-source datasets to inform spatial targeting of active case detection for skin NTDs at district and lower administrative levels, focusing on Ghana as an example. We present a conceptual framework, that includes contributing factors, for combined risk of endemicity and under-detection, use an evidence consensus framework to systematically combine and evaluate different data sources, and rank administrative units according to their suitability for targeting according to multiple criteria. We also present our results in a format which we believe is accessible to target users, and does not require specialist software for viewing and manipulation. The output is a tool designed to guide decision makers to best deliver integrated active case detection for skin NTDs.

## Materials and methods

### Ethics statement

All data used in this study are publicly available and fully anonymised. For DHS data, both the primary and senior authors obtained permission to access and use the anonymised, community-level data.

As this study did not involve human participants and relied solely on model-based predictions, ethical review was not required.

### Study design

This was an analysis of secondary data using an evidence consensus approach to summarise and synthesise a variety of spatial datasets.

### Study area

We analysed data for Ghana, a West African country with a predicted population of 34.8 million in 2024 [30]. The country is divided into sixteen regions and 260 districts (as of 2019), which were the main unit of our analysis. We obtained the shapefile used for the basemap of this analysis from the Humanitarian Data Exchange [31].

In Ghana the skin NTDs known to be endemic are yaws [32], BU [33], LF [34], scabies [35], CL [36], onchocerciasis [4] and leprosy [18,37]. Multiple national level control programs exist for skin NTDs in Ghana, including the National Buruli Ulcer Control and Yaws Eradication Program, and the National Leprosy Control Program, though integration across these programs is promoted by the national NTD Masterplan [38].

## Conceptual framework

We developed a conceptual framework consisting of five domains, representing broad criteria which could be used for targeting active case searches and case management interventions for skin NTDs. The five domains were 1) evidence for endemicity, 2) population vulnerability to disease, 3) accessibility of health services, 4) availability of health services, and 5) risk factors for poor clinical outcomes (Fig 1).

We then conducted a rapid literature [39] —an approach that follows the principles of a systematic review but uses streamlined methods to allow for a quicker synthesis of the available evidence — review to identify spatially-varying risk factors for each of the five domains and sought spatial datasets which could act as direct indicators or suitable proxies for them. We recorded the sources of selected datasets and those factors for which we were unable to identify suitable indicators or proxies.

## Domains

**Evidence for endemicity.** The first domain was intended to represent the likelihood of target diseases occurring in each administrative unit, and theoretically could be based on any or a combination of: cases recorded through surveillance; local expert opinion (e.g., anecdotal reports of cases not officially recorded); cases reported in published literature; or prevalence or suitability predictions. In the case of this study, predicted BU suitability and predicted LF antigenaemia were utilised. In this context, 'suitability' denotes the availability of environmental conditions which best characterise locations with occurrence of Buruli ulcer cases, derived from a modelling study.

**Population vulnerability to disease.** The second domain covered risk factors for the diseases, either shared or disease specific. For example, low housing quality was considered to be a shared risk factor across several target

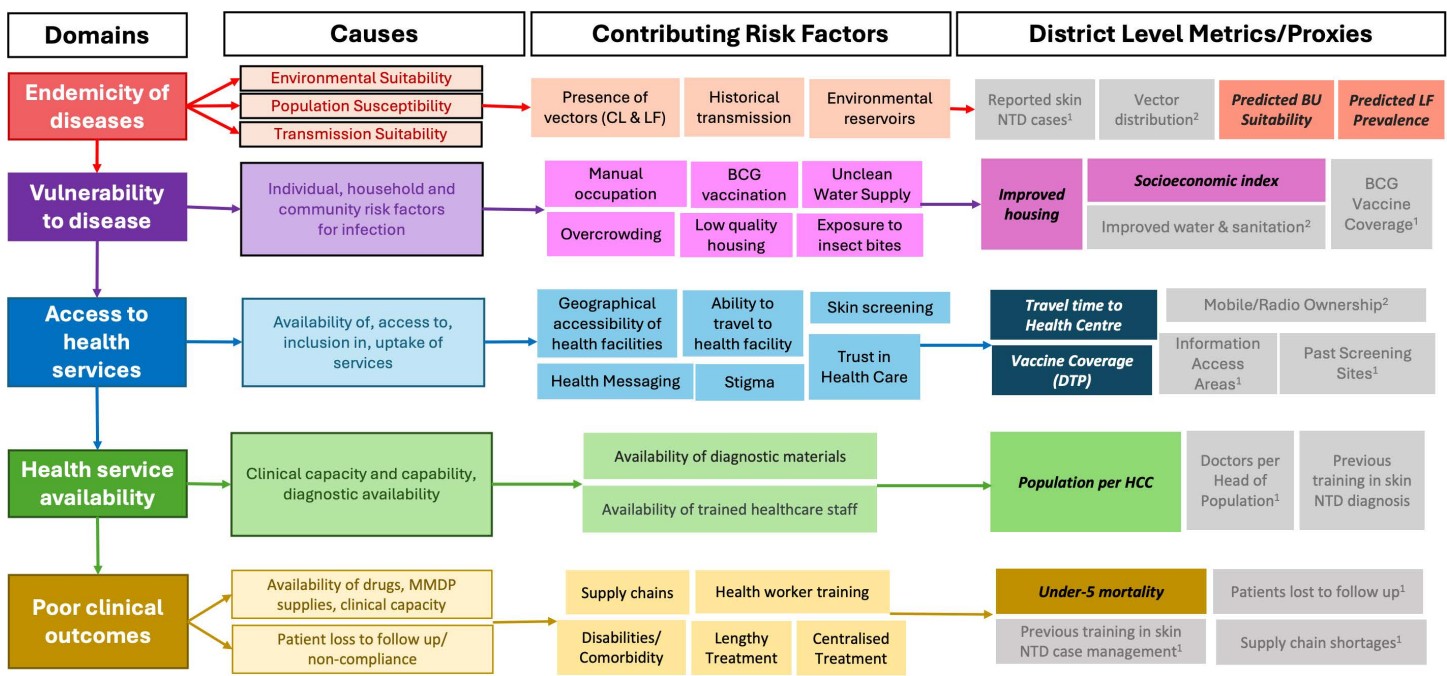

**Fig 1. Conceptual framework.** BCG – Bacillus Calmette-Guérin (vaccine), BU – Buruli ulcer, DTP – Diphtheria, tetanus, and pertussis (vaccine), HCC – Health care centre, HH – Household, LF – Lymphatic filariasis, SE – Socioeconomic, U5M – Under-5 mortality. *Grey boxes show indicators which we sought but did not use for the following reasons: 1 = not publicly available; 2 = overlaps extensively with another indicator (predictions use similar data sources or one is as a predictor of the other).

diseases, considering that well-built houses may provide protection against vectors like mosquitoes, resulting in a lower risk of LF infection[38], and would be less likely to be overcrowded, reducing the risk of leprosy and yaws transmission [40]. Socioeconomic-status was intended to act as a general marker for and living conditions and vulnerability to disease.

**Accessibility of health services.** The third domain was intended to represent under-detection as a result of cases not being able to access health services. Longer travel times to health care centres can act as a barrier to seeking medical attention. We also considered the coverage of essential health services, such as childhood immunisations, as an indicator of health system strength and population uptake of primary health services.

**Availability of health services.** This domain was intended to represent under-detection as a result of limited diagnostic capacity at health facility level. We considered that this would reflect the availability of diagnostic tests, diagnostic capabilities of the health workforce, and healthcare staff and facilities per head of population.

**Risk factors for poor clinical outcomes.** This domain was included in order to prioritise selection of areas in which cases may have been identified but did not achieve complete clinical cure due to loss to follow-up or shortages of health staff or treatment supplies. We identified under-five-mortality as a spatially varying metric which could act as a proxy for health system performance.

**Data sources and processing.** We assembled indicator and proxy variables from publicly available datasets in three common formats: raster files (described previously as maps made up of evenly spaced grid cells, each containing a value such as predicted prevalence), shapefiles (digital map files that store the boundaries and shapes of geographic areas such as districts), and tabular data (spreadsheets containing numerical or categorical information). All maps were converted into the same format and scale so they could be compared district by district. For example, we reprojected the spatial data into a common coordinate reference system (EPSG:4326, WGS84) and adjusted the size of each reporting unit by resampling to a uniform resolution of 5 × 5 km. Details of their sources and are shown and cited in Table 1.

## Data sources and processing

**Selection of target districts.** To streamline analysis, we filtered the 260 districts (as of 2019 – Guan district introduced in 2021) to those where co-endemicity occurred. To achieve this we used two metrics, predicted suitability for BU and LF prevalence estimates. These two metrics were used due to the ready availability of the datasets and shared endemicity in Ghana; using both could indicate the co-endemicity of other skin NTDs. Predicted suitability for BU was obtained in raster format (a continuous gridded spatial dataset) from a modelling study which used recorded occurrences of the disease and environmental covariates to predict the distribution of BU across Africa [19]. LF prevalence estimates for Ghana were obtained in raster format from a geostatistical and mathematical model [22]. We used antigenaemia estimates from a 2015 paper by Moraga *et al* [22], as prevalence from approximately 10 years ago was considered a more suitable predictor of current morbidity burden than recent estimates. [50]For each district we calculated the proportion of total area predicted to be suitable for BU [19] and the mean predicted prevalence of LF antigenemia [22]. We excluded all districts predicted fully unsuitable for BU. For predicted LF prevalence, no districts were 0%, therefore mean district prevalence was divided into quintiles and districts in the bottom quintile (with predicted prevalence of 5% or less) were excluded. Districts in the bottom quintile of predicted LF prevalence (≤5%) were excluded to focus the analysis on locations with meaningful transmission potential. Including very low-prevalence districts would have increased the number of areas with minimal expected cases, reducing the sensitivity of the analysis to identify high-priority districts for active case detection.

**Evidence for endemicity.** To achieve this we used two metrics, predicted suitability for BU [19] and LF [22] prevalence estimates, which were the only two readily available sources of information on the endemicity of the target diseases (Table 2). We considered both of these indicators together to identify in which districts both predicted BU suitability and predicted LF prevalence where present, in order to recommend suitable locations for integrated interventions. We classified districts into five levels (one – five), the first representing no predicted suitability for BU, and the other four levels classified at breaks of 0.25, 0.5 and 0.75 (representing the proportion of district area predicted suitable). We also

 

**Table 1. Indicators and proxies used for the evidence consensus.**

| Domain | Metric/Proxy | Sources (original CRS: Resolution) |
|---|---|---|
| **Evidence for endemicity** | Predicted BU Suitability | Simpson et al 2021 [19] (EPSG 3857: 5km) |
| | Reported BU cases | *Not publicly available* |
| | Predicted LF antigenaemia | Moraga et al 2015 [22] (EPSG 3857: 1km) |
| | Reported LF lymphedema cases | *Not publicly available* |
| **Population Vulnerability to Disease** | Predicted Improved Housing Prevalence | Malaria Atlas Project** [41] (EPSG 4326: 5km) |
| | Relative Wealth Index | Demographic and Health Surveys Program website in the form of their 2017 Special Survey [42] (EPSG 4326) |
| | Age Distribution | United Nations Population Fund *via* the Human Data Exchange [42]. |
| | Mobile Phone Ownership | Demographic and Health Surveys Program website in the form of their 2017 Special Survey [42] (EPSG 4326) |
| **Accessibility of health services** | Information Access Areas* | *Not publicly available* |
| | Past Intervention Sites | *Not publicly available* |
| | Past Survey Sites | *Not publicly available* |
| | Travel Time to HCC | A spatial database of health facilities managed by the public health sector in sub Saharan Africa [43], Open Street Map [44] (EPSG 4326), Esri - Sentinel-2 Land Cover Explorer [45] (EPSG:32630: 10m, EPSG:32631:10m) |
| | Vaccination Coverage (DPT) | Institute for Health Metrics and Evaluation (IHME) [46]. (EPSG 4326: 1km) |
| | Pop Using Traditional Medicine | *Data Unknown* |
| **Availability of health services** | Population per HCC | A spatial database of health facilities managed by the public health sector in sub Saharan Africa [43], Open Street Map [44], WorldPop [47] (EPSG 4326: 1km). |
| | % Rural Population | Esri - Sentinel-2 Land Cover Explorer [45] (EPSG:32630: 10m, EPSG:32631: 10m), WorldPop [47] (EPSG 4326: 1km). |
| | Doctors per Head of Population | *Publicly Unavailable* |
| **Risk factors for poor clinical outcomes** | Under 5 Mortality | Institute for Health Metrics and Evaluation (IHME) [48] (EPSG 4326: 5km). |
| | Doctor Training Level | *Dataset Unavailable* |
| | HCC Types in District | A spatial database of health facilities managed by the public health sector in sub Saharan Africa [43] (EPSG 4326), Open Street Map [44] (EPSG 4326) |
| | Healthcare Readiness | Ghana Round 6 SQ Survey *via* the Performance Monitoring for Action Website [49]. |
| | Availability of Essential Medicines | Ghana Round 6 SQ Survey *via* the Performance Monitoring for Action Website [49]. |

*Information Access Area's such as; Health Centres, Centres for education, Community radio services, Mobile information services, Public Libraries, Community Centre. **Information on this reference's link can be accessed by selecting the drop down menus on the left of the screen and proceeding to press the download button.

classified districts into quantiles of predicted LF prevalence, two representing the lowest quantile and five representing the quantile with the highest predicted LF prevalence). The scores across predicted BU suitability and predicted LF prevalence were combined to give a score out of ten (Table 3).

**Population vulnerability to disease.** We used two indicators to represent this domain. We used a raster dataset of the predicted prevalence of improved housing in 2015, obtained from the Malaria Atlas Project [41]. We calculated the mean prevalence for each district. We also obtained data from the Ghana 2017 Special Survey conducted through the Demographic

**Table 2. Summary of indicators used in evidence consensus.**

| Domain | Metric | Quintile | Count | Mean | Min | Max |
|---|---|---|---|---|---|---|
| Evidence for Endemicity | Predicted BU Suitability | 5 | 22 | 0.973 | 0.916 | 1 |
| | | 4 | 21 | 0.764 | 0.59 | 0.904 |
| | | 3 | 22 | 0.411 | 0.268 | 0.58 |
| | | 2 | 21 | 0.145 | 0.09 | 0.266 |
| | | 1 | 22 | 0.032 | 0.001 | 0.089 |
| | Predicted LF Prevalence | 5 | 22 | 0.251 | 0.172 | 0.362 |
| | | 4 | 21 | 0.144 | 0.121 | 0.171 |
| | | 3 | 22 | 0.092 | 0.077 | 0.113 |
| | | 2 | 21 | 0.06 | 0.051 | 0.074 |
| | | 1 | 22 | 0.039 | 0.029 | 0.049 |
| Vulnerability to Disease | Predicted Improved Housing | 5 | 22 | 0.085 | 0.042 | 0.115 |
| | | 4 | 21 | 0.131 | 0.118 | 0.147 |
| | | 3 | 23 | 0.17 | 0.148 | 0.192 |
| | | 2 | 20 | 0.225 | 0.196 | 0.254 |
| | | 1 | 22 | 0.293 | 0.256 | 0.376 |
| | Relative Wealth Index | 5 | 23 | 2.349 | 1.55 | 2.724 |
| | | 4 | 20 | 3.028 | 2.729 | 3.213 |
| | | 3 | 22 | 3.334 | 3.237 | 3.43 |
| | | 2 | 21 | 3.594 | 3.444 | 3.754 |
| | | 1 | 22 | 4.05 | 3.758 | 4.739 |
| Access to Health Services | Travel time to HCC | 5 | 22 | 1.784 | 1.379 | 3.16 |
| | | 4 | 21 | 1.22 | 1.061 | 1.373 |
| | | 3 | 22 | 0.886 | 0.704 | 1.059 |
| | | 2 | 21 | 0.496 | 0.222 | 0.702 |
| | | 1 | 22 | 0.065 | 0 | 0.219 |
| | Vaccination Coverage (DPT) | 5 | 23 | 0.82 | 0.719 | 0.872 |
| | | 4 | 20 | 0.883 | 0.874 | 0.892 |
| | | 3 | 22 | 0.907 | 0.894 | 0.917 |
| | | 2 | 21 | 0.928 | 0.918 | 0.934 |
| | | 1 | 22 | 0.942 | 0.935 | 0.952 |
| Availability of Health Services | Population per HCC | 5 | 22 | 27805 | 17874 | 49501 |
| | | 4 | 21 | 14253 | 11569 | 17336 |
| | | 3 | 22 | 10021 | 8476 | 11090 |
| | | 2 | 21 | 7465 | 6528 | 8308 |
| | | 1 | 22 | 4335 | 2182 | 6479 |
| Risk Factors for Poor Clinical Outcomes | Under 5 Mortality | 5 | 22 | 0.109 | 0.106 | 0.118 |
| | | 4 | 19 | 0.102 | 0.099 | 0.105 |
| | | 3 | 21 | 0.096 | 0.095 | 0.098 |
| | | 2 | 22 | 0.093 | 0.091 | 0.094 |
| | | 1 | 24 | 0.082 | 0.073 | 0.09 |

and Health Surveys (DHS) Program, derived from the Ghana DHS Phase 7 household dataset (GHHH7JFL) [42]. This dataset included 26,324 georeferenced households in 900 different clusters [42]. A relative wealth index, adjusted for rurality, had been calculated from a variety of indicators and classified into quintiles. DHS surveys use a standardised core methodology across participating countries to ensure comparability, but the exact indicators collected can vary by country and by survey year

**Table 3. Districts in the top quintile of total score across domains, and indicators of endemicity.**

| Region | District | Total | D1 | D2 | D3 | D4 | D5 | BU Indicator | LF Indicator |
|---|---|---|---|---|---|---|---|---|---|
| BONO EAST | PRU EAST | 40 | 3 | 10 | 7 | 10 | 10 | 0.99 | 0.17 |
| WESTERN | SHAMA | 40 | 10 | 7 | 5 | 10 | 8 | 0.43 | 0.09 |
| WESTERN | NZEMA EAST | 39 | 8 | 7 | 10 | 4 | 10 | 0.11 | 0.15 |
| CENTRAL | ASSIN SOUTH | 38 | 5 | 6 | 7 | 10 | 10 | 0.27 | 0.24 |
| CENTRAL | AWUTU SENYA EAST | 38 | 10 | 6 | 4 | 8 | 10 | 1 | 0.36 |
| CENTRAL | EKUMFI | 38 | 10 | 7 | 3 | 8 | 10 | 1 | 0.27 |
| WESTERN | MPOHOR | 38 | 8 | 8 | 8 | 4 | 10 | 1 | 0.17 |
| EASTERN | UPPER WEST AKIM | 38 | 6 | 8 | 6 | 10 | 8 | 0.48 | 0.34 |
| WESTERN | WASSA EAST | 38 | 6 | 8 | 10 | 4 | 10 | 0.01 | 0.14 |
| GREATER ACCRA | GA SOUTH MUNICIPAL | 37 | 9 | 6 | 4 | 10 | 8 | 0.02 | 0.1 |
| CENTRAL | GOMOA WEST | 37 | 8 | 8 | 5 | 6 | 10 | 0.45 | 0.19 |
| CENTRAL | KOMENDA-EDINA-EGUAFO-ABIREM MUNICIPAL | 37 | 9 | 6 | 4 | 8 | 10 | 0.62 | 0.21 |
| WESTERN | PRESTEA/HUNI VALLEY | 37 | 4 | 9 | 10 | 6 | 8 | 0.9 | 0.17 |
| AHAFO | TANO NORTH MUNICIPAL | 37 | 6 | 8 | 7 | 10 | 6 | 0.02 | 0.06 |
| WESTERN | WASSA AMENFI EAST | 37 | 3 | 8 | 10 | 8 | 8 | 0.03 | 0.06 |
| EASTERN | AYENSUANO | 36 | 7 | 9 | 6 | 8 | 6 | 0.27 | 0.13 |
| WESTERN | EFFIA KWESIMINTSIM MUNICIPAL | 36 | 9 | 3 | 6 | 10 | 8 | 0.17 | 0.17 |

D1-5 = Domains 1–5. D1 = Indicators of Endemicity; D2 = Population Vulnerability to Disease; D3 = Low Access to Health Services; D4 = Low Availability of Health Services; D5 = Risk Factors for Poor Clinical Outcomes. BU Indicator = Proportion of district area predicted suitable for Buruli ulcer; LF Indicator = Predicted prevalence of lymphatic filariasis.

depending on national priorities and resources. For this analysis, we selected DHS indicators that were available for Ghana and that are commonly collected in many other countries, recognising that adaptation would be needed where these variables are unavailable elsewhere. We aggregated households to districts and calculated the mean relative wealth index quintile of each district. Of the 260 districts in Ghana, eleven were not represented by household clusters. For these districts with missing values, we imputed the mean relative wealth index quintile of their neighbouring districts.

**Accessibility of health services.** For this domain we used an indicator representing accessibility to health facilities and a proxy variable representing inclusion within health programs. For the first, we modelled travel times to recorded health facilities using the R package *gdistance* [51]. We obtained coordinates of healthcare centres recorded on Open Street Map [44] and from a dataset provided by the Malaria Atlas Project [43]. These datasets were checked and deduplicated by removing facilities represented by multiple points or within 500m of another facility. Travel times were estimated based on movement speeds determined by the road network and landcover. A shapefile representing the road networks of Ghana was obtained from Open Street Map [44] via the Humanitarian Data Exchange website and was converted to a raster dataset [52]. Raster datasets of land usage were obtained from the ArcGIS website using the Esri Sentinel-2 Land Cover Explorer App [45]. Assumed movement speeds were assigned to different categories of road type and surface and to different landcover types, and the two datasets were then combined to provide a single dataset representing travel speed across each grid cell. This was used to calculate "accumulated costs" or the total travel time from each cell to a health facility following the route of lowest accumulated costs. We calculated the mean estimated travel time for each district.

To represent inclusion within health programs we used a raster dataset representing coverage of diphtheria-pertussis-tetanus vaccine for Africa (2016) from the Institute for Health Metrics and Evaluation (IHME) [46]. We calculated mean coverage for each district.

**Availability of health services.** This domain was represented by the population per health care centre. We used UN adjusted population data from the WorldPop Project [47] to estimate total population for each district and the consolidated dataset of HCCs described above [43,44]. Population per health care centre was then calculated. Healthcare centres include health posts (the first level of primary care in rural areas); health centres and clinics; or hospitals.

**Risk factors for poor clinical outcomes.** As a proxy for this domain, we used estimates for the probability of death before 5 years old from 2000-2017, obtained as raster files from the Institute for Health Metrics and Evaluation (IHME) [48]. We calculated the mean probability across years and then calculated the mean for each district.

All data processing was done using R Statistical Software, primarily using the *terra* [53] and *sf* packages [50,54,55].

**Evidence synthesis.** As the primary aim was to prioritise districts for integrated active case detection of skin NTDs, we restricted analyses to districts with plausible co-occurrence of BU and LF. Co-endemic districts were therefore defined as those with any area predicted suitable for BU and a mean predicted LF antigen prevalence of at least 5%. This threshold was chosen to exclude districts with low predicted LF prevalence where undiagnosed morbidity cases would be rare [56]. This pragmatic definition was selected to focus the framework on districts most likely to benefit from integrated case detection rather than to provide a formal epidemiological definition of co-endemicity.

For the remaining "co-endemic" districts, each of the assembled indicator datasets was classified into quintiles, and each level was given an indicator score from 1 (low prioritisation of active case finding) to 5 (highest prioritisation of active case finding). For each district, indicator scores were summed within domains to give domain scores. To ensure that all domains carried equal weight, regardless of the number of component indicators, we adjusted the domain scores by linear scaling. Specifically, the scores from the "Availability of Health Services" and "Risk Factors for Poor Clinical Outcomes" domains were doubled to bring them to an equivalent maximum score of 10, aligning with the other domains.

**Visualisation.** R Markdown [57] was used to create a dashboard with the data displayed in visual and tabular form. The leaflet package was used to map district-level domain and total scores on an interactive map, with a separate layer for each domain and a 6th for the total quintile score for each district. The raw data and indicator scores for each district were displayed in separate tabs for aiding decision making. A user guide was included on the dashboard.

## Results

### Indicator and domain scores

**Evidence for endemicity.** The final number of districts identified as co-endemic for BU and LF was 108. These 108 districts were primarily located in the south of Ghana, excluding the south-east. The majority of districts in the north had no predicted suitability for BU and were therefore removed (S1 Fig). Districts of high predicted suitability for BU were clustered around the central-south of the country and along the coastline. The range of predicted BU suitability was 0 – 100% and 0.001 – 100% when excluding those districts which did not contain predicted suitability for BU (Table 2).

The range of predicted LF prevalence, as a proportion of district population, was 0.029-0.362 (Table 2). Districts with higher predicted LF were located in the mid-west and central-south of the country, while most districts in the selected 108 with lower predicted antigenaemia were in the south east excluding coastal districts (S1 Fig).

Adjusted domain scores ranged from two to ten. Generally, there was little overlap between districts predicted highly suitable for BU and high prevalence for LF, and the maximum score was met by only four districts, which formed a cluster in the south (Shama, Mfantseman Municipal, Ekumfi and Awutu Senya East). Districts surrounding this cluster also scored relatively high in this domain (scores of 7–10), as did the districts along most of the coast in the south.

**Population vulnerability to disease.** The mean predicted prevalence of improved housing ranged from 0.042- 0.376 (Table 2 and Fig 2). Districts were classified into five approximately equal groups. Districts with the highest indicator scores (representing lower levels of predicted improved house prevalence) were located in the north of Ghana and outside the main cities. In the final 108 districts high levels of improved housing were seen in the south-east along the

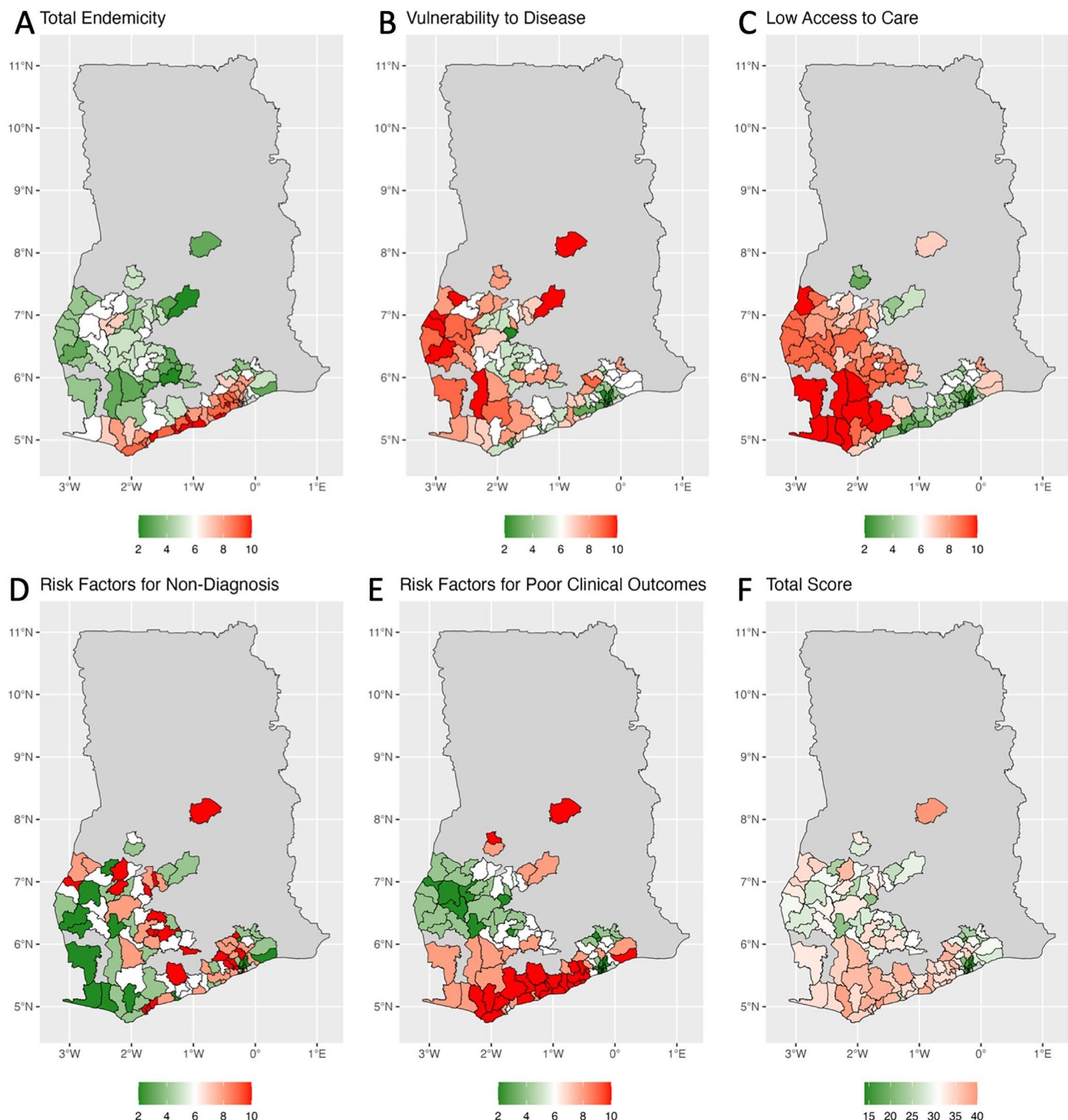

**Fig 2. Adjusted domain scores. A)** Indicators of Endemicity **B)** Vulnerability to Disease **C)** Low Access to Care; **D)** Risk Factors For Non-diagnosis; **E)** Risk Factors for Poor Clinical Outcomes. 3F: Total Prioritisation Scores. Map shapefiles obtained from the Humanitarian Data Exchange (HDX), Ghana – Subnational Administrative Boundaries, 2023, under a CC BY-IGO license (https://data.humdata.org/dataset/cod-ab-gha) [31].

coast (S1C Fig). Districts with higher indicator scores for relative wealth index (lower wealth) were mainly in the east and west of the country and outside the major cities in Ghana. In the Final 108 districts, high and low scores were dispersed evenly (S1D Fig). There was extensive overlap between districts with lower prevalence of improved housing and lower relative wealth. As a result, 7 districts were assigned the highest score within this domain. These districts were all located in the west of the country, predominantly rural areas.

**Accessibility of health services.** Travel time to the nearest health centre ranged from 0.000 to 3.16 hours (Table 2 and Fig 2). Districts with the shortest travel time to health care centres were located predominantly in the centre of the country and along the coast including cities such as Accra and Kumasi in the South. Districts with the longest travel times to health centres were spread across rural areas in the south-west and centre-east of the country (S1E Fig). The lowest district-level DPT coverage was 0.719 and the highest was 0.952.

Most districts with lowest DPT coverage were located in the south-west and north-east, while there were clusters of districts with higher coverage in the centre, south and east of the country (S1F Fig).

There was overlap of high travel times to health centres and low DPT coverage in the south-west of the country. Scores ranged from two to ten, with 9 assigned ten. These 9 were mostly located in the south-west.

**Availability of health services.** This domain was represented by population per health care centre, using the 3,322 health care centres as in the Accessibility domain. The highest population per health care centre (representing lowest availability of health services) was 49,501 people, while the lowest was 4,335 (Table 2 and Fig 2). Districts with high population per health centre were distributed throughout the country, but with clusters in the centre and south (S1G Fig). There were clusters of districts with low population per health centre throughout the country but in the final 108 districts they are located predominantly in the south-west. This was the only indicator within this domain, so domain scores reflect individual scores for this indicator.

**Risk factors for poor clinical outcomes.** Under 5 mortality (U5M) was used as a proxy for risk factors for poor clinical outcomes. The district-level mean U5M ranged from 0.073-0.118 (Table 2 and Fig 2). Districts with high U5M were primarily located along the coastline, while districts with lower levels of U5M were mainly in the middle of the country and around Accra on the south coast (S1H Fig). As with the previous domain, this was the only indicator.

**Datasets unavailable.** We were unable to obtain data on or indicators/proxies of: reported NTD cases (Endemicity domain); BCG vaccine coverage (Vulnerability to Disease domain); mobile phone/radio ownership, information access areas, past intervention (Access to Health Services domain); doctors per capita (Availability of Health Services domain) (Table 2).

**Total scores.** Total scores for each district are shown Fig 2F and Table 3. Total scores ranged from 14-40 out of a possible maximum of 50. There were 17 districts in the highest quintile of Total Score (Scores of greater than 36/50). Districts at the upper end of this quintile were predominantly in the south-west and central-east. In the highest quintile of Total Score, scoring of LF and BU ranges from 1-5 with the total scores in Evidence for Endemicity ranging from 3-10. 13/17 scored 4 or 5/5 for predicted LF prevalence while 6/17 scored 4 or 5/5 for predicted BU suitability. These districts were mostly located in the south along the coast. The lowest scoring districts were all in the south-east of the country, with clusters around Accra on the southern cost, in the south-east interior, and on the eastern border.

**Dashboard.** The final Dashboard summarising the output is available publicly on Rpubs at the link below.
[http://rpubs.com/Williamjoneswarner/skin_ntd](http://rpubs.com/Williamjoneswarner/skin_ntd)

## Discussion

This evidence consensus framework demonstrates a workflow by which decision makers in countries with suspected or known endemicity of skin NTDs can use open-source data to help guide selection of locations for active case detection. In brief, we identified five domains that could influence where undiagnosed cases might occur: evidence for endemicity, population vulnerability, accessibility of health services, availability of health services, and risk factors

for poor clinical outcomes. Second, we located open-source datasets that could provide information for each domain. Third, we summarised each dataset for every district in Ghana. Fourth, we converted these values into comparable scores so that they could be directly compared across different types of information. Finally, we combined the domain scores into a single total score for each district, which allowed us to rank them according to priority for active case detection.

We had the overall aim of creating a tool to help in the process of identifying undiagnosed cases of skin NTDs, by guiding active case detection efforts in Ghana, with a predominant focus on BU and LF. We created a conceptual framework to help guide us on factors that would impact the potential for a district to have increased risk of individuals with undiagnosed skin NTDs, based on evidence from environmental modelling and socio-economic factors. The five domains outlined in Fig 1 demonstrate what we perceived as the 5 main influencing factors. To quantify these risks, we collated open-source spatial datasets that would act as indicators or proxies for these risks. The use of open-source datasets, typically in the form of raster files, enables flexible analysis because they store information as a regular grid of values (e.g., predicted prevalence or travel time) that can be overlaid with shapefiles showing district boundaries. This makes it straightforward to calculate summary statistics—such as the average value within each district—regardless of the size or shape of the district, and to repeat the same process for different geographic areas or administrative levels. The conversion of raw data at the acquisition stage and conversion into easily interpretable quintiles allows users to quickly assess and analyse the data to maximise the efficiency of the interpretation.

Importantly, the results are presented in an interactive online dashboard that does not require any GIS or R programming skills to use. Decision-makers can simply open it in a web browser, view maps and tables, click on districts to see their scores, and explore each domain without installing specialist software. This ensures that the tool can be readily accessed and used by a wide range of stakeholders, including those without technical mapping experience. The same approach can be adapted for other diseases, countries, or geographic levels if suitable datasets are available. Even for users who cannot run R code or use GIS software, the final dashboard outputs can be shared as an interactive web tool, allowing exploration of results without any specialist technical skills.

We next consider the current application of this framework for BU and LF. As this framework currently stands, it can be used for the prioritisation of districts for active case detection of LF and BU. Due to the 5 different domains built from 8 different indicators, users may prioritise which domain and indicator they feel is best suited for choosing districts to target. From this, users can then sort on secondary and tertiary domains or individual indicators that would help fine-tune the choice.

We chose to prioritise districts based on their overall sum scores across the five domains, refining our analysis by examining the Indicators of Endemicity domain for predicted BU suitability and LF prevalence to align with our goals. This process identified 108 districts potentially suitable and possibly endemic for both LF and BU, of which 17 were in the highest scoring quintile (scores ≥ 36), making them strong candidates for active case detection of skin NTDs.

Although WHO elimination guidelines for lymphatic filariasis use very low prevalence thresholds (for example, < 1% microfilaria or <2% antigenaemia) to determine when transmission is no longer sustainable and mass drug administration can be stopped, districts with predicted prevalence of 5% or less were excluded from our analysis. At such low predicted prevalence, the likelihood of ongoing transmission and undiagnosed cases is minimal compared with higher-prevalence areas. Including these low-prevalence districts could have introduced additional noise, reducing the ability of the framework to detect and prioritise districts with meaningful transmission potential [56].

Among these, three districts scored above 39 —Pru East, Shama, and Nzema East—scoring 40, 40, and 39, respectively, marking them as the highest-ranking overall (Table 3). However, their suitability for BU was varied from 0.11 to 0.99, suggesting total high score alone might not be the best indicator for BU-specific detection efforts. Six districts, including Shama, Awutu Senya East, Ekumfi, Komenda-Edina-Eguafo-Abirem Municipal, Effia Kwesimintsim Municipal, and Ga

South Municipal, scored highly (4 or 5) for both BU suitability and LF prevalence, indicating higher disease levels. Notably, BU suitability coverage within these districts varied significantly, ranging from 62.1% to 100%.

Given the limited resources available for active case detection, focusing on districts with both high overall scores and a substantial proportion of BU-suitable areas may optimize detection efforts. While Pru East, Shama, and Nzema East stand out for their total scores, districts like Awutu Senya East and Ekumfi may be more practical targets for BU-focused detection due to their greater BU suitability than Pru East and Nzema East.

They key strength of this study is the flexibility of the workflow, which would allow users with a basic knowledge of R programming and/or GIS skills to adapt it according to local needs and disease endemicity. For example, the framework could be applied to different countries, and/or adapted to cover different diseases, different indicators, and to assign different weights to indicators depending on programmatic priorities. In practice, this workflow could follow a simple sequence: (1) identify the target disease(s); (2) decide on the most relevant domains influencing case detection (e.g., endemicity, vulnerability, access, availability, clinical outcomes); (3) identify open-source datasets or suitable proxies for each domain; (4) summarise these datasets at the desired geographic level; (5) convert them into comparable scores; and (6) combine these scores to produce a ranked list of priority areas for intervention.

There are certain limitations which should be considered if this tool were to be used to guide programmatic activities. This framework was produced using a shapefile of district level administration zones, which vary in size and are extremely heterogenous in certain indicators, such as distance to health care centre. This means that the total score allocated to a district may mask smaller areas with higher scores across all indicators. If a subdistrict shapefile were available, using this as the unit of analysis would increase the granularity of evaluation and the efficiency of resource allocation. The flexibility of the workflow used means that this would be straightforward to implement if required.

Further limitations are introduced by the choice and availability of indicators. We only included endemicity indicators for LF and BU as predictions for their prevalence/ suitability were available, which is not the case for other endemic skin NTDs which could be controlled by case detection and integrated skin care programs (notably leprosy and yaws). These predictions were used as they are open-source, but are associated with a degree of uncertainty, especially those for BU given the limited understanding of risk factors associated with this disease. If available, surveillance data would be a useful additional indicator of endemicity of the range of endemic skin NTDs. Other datasets which would have been included if available included doctors per head of population, BCG vaccination coverage and past intervention sites. While some of these datasets may available to governmental health departments, they cannot always be shared, and so we decided to use alternative indicators or proxies.

Several limitations related to data availability and representativeness should be considered when interpreting these findings. District-level wealth estimates derived from DHS household data should be interpreted with caution, as the DHS is designed to produce nationally and regionally representative estimates and is stratified by urban and rural residence rather than by district. Consequently, the relative wealth index values used here, derived from the Ghana DHS Phase 7 household dataset, should be viewed as approximate indicators of relative socioeconomic vulnerability rather than precise district-level estimates. Because the DHS sampling design is not intended to be representative at the district level, the households sampled within a given district may not fully reflect the socioeconomic characteristics of the district population as a whole, particularly in districts with few sampled clusters [42].

In addition, some open-source datasets contained missing values for specific districts, namely the DHS dataset. To enable inclusion of all districts, we applied single mean imputation, replacing missing values with the mean of neighbouring districts. While this pragmatic approach allowed complete district coverage, it does not account for uncertainty in the imputed values and may underestimate true variability compared with more robust methods such as multiple imputation by chained equations. These limitations were acknowledged and considered acceptable given the intended use of the framework as a decision-support tool for a working group of experts familiar with the local context. Importantly, individual indicators contribute only one component of a broader, multi-domain assessment rather than serving as standalone

determinants. As such, scores and rankings for districts with imputed or sparsely sampled data should be interpreted cautiously, and future applications could incorporate multiple imputation approaches to better quantify uncertainty and strengthen the robustness of prioritisation [58].

An important limitation of this study is the absence of validation using empirical data on BU and LF at the district level. Given that our primary aim was to identify areas with potential undiagnosed disease, robust validation would require data from population-based prevalence surveys across districts with different priority scores, which were beyond the scope of this study. Routine surveillance data were not considered an appropriate benchmark for validation of true disease burden, as they may be more reflective of surveillance sensitivity and health system access than underlying transmission intensity or unmet need. A validation study for LF demonstrated that routine surveillance underestimates lymphoedema by approximately 80% compared with community-based surveys [59]. Similar limitations apply to Buruli ulcer, where hospital- and clinic-based reporting is known to capture only a fraction of community cases due to delayed care-seeking, misdiagnosis, and limited access to diagnostic services [60,61]. As a result, observed case counts are more reflective of surveillance sensitivity and health system access than underlying transmission intensity or unmet need. Given that the primary aim of this framework is to identify areas with potential undiagnosed disease rather than areas with high reported case numbers, routine surveillance data were not used for internal validation. Robust validation of the framework will ultimately require targeted morbidity surveys for LF and active case-finding surveys for BU in prioritised districts, which were beyond the scope of the present study but represent an important direction for future work.

Another limitation of this tool is that while it may help guide decision makers towards a location for active case detection strategies, it does not take into account the feasibility of these districts for surveying, the logistical ease of implementing surveys in these areas, the acceptance of the local population for community engagement, sustainability and scalability of surveying to certain areas or integration with other public health interventions, all of which need to be considered.

## Conclusions

This evidence-based framework has demonstrated a method by which publicly available datasets can be combined into an easily interpretable format to support decision-makers designing skin NTD control strategies. From our assessment, active case detection strategies for BU and LF should be prioritised in districts scoring highly on our combined total. Additionally, districts with high predicted prevalence of LF or suitability for BU should be considered even if they scored lower across other domains. In summary, this evidence-based framework not only simplifies the integration of public datasets for enhanced clarity and interoperability but also strategically guides the implementation of active case detection for BU and LF, ensuring comprehensive coverage and addressing gaps in current knowledge and geographical prediction.

### Glossary of key technical terms

**Coordinate Reference System (CRS)** – A framework that defines how geographic coordinates (latitude and longitude) relate to positions on the Earth's surface.

**Domain** – A broad category used in the framework to group related factors affecting disease risk or detection (e.g., endemicity, access to health care).

**Endemicity** – The extent to which a disease is consistently present in a particular geographic area.

**EPSG:4326 (WGS84)** – A commonly used CRS based on latitude and longitude, compatible with most mapping tools.

**Evidence Consensus Framework** – A structured method for combining and weighting multiple sources of information, such as standardised data on demographics or infection, to guide decision-making.

**Geographic Information System (GIS)** – Software for storing, analysing, and visualising spatial data (e.g., QGIS, ArcGIS).

**Indicator** – A specific measurable variable representing a factor within a domain (e.g., travel time to the nearest health facility as an indicator of accessibility).

**Interactive dashboard** – A web-based tool that allows users to view and interact with data through maps, tables, and charts, without specialist software.

**Open-source data** – Data freely available for use, reuse, and distribution, often without licensing restrictions.

**Proxy variable** – A variable used to indirectly measure something when a direct measurement is not available (e.g., poverty level as a proxy for vulnerability).

**Quintile** – A method of dividing data into five equal groups, each containing 20% of the values, for easier comparison.

**Ranking** – Ordering districts from highest to lowest priority based on combined scores from all domains.

**Raster file** – A type of digital map made up of evenly spaced grid cells, each containing a value such as predicted prevalence, rainfall, or travel time.

**Shapefile** – A digital file format that stores the shapes and boundaries of geographic features, such as administrative districts or roads.

**Spatial dataset** – Any dataset that contains information linked to specific locations or geographic coordinates.

**Standardisation (of scores)** – The process of adjusting values from different datasets so they can be compared fairly.

## Supporting information

**S1 Fig. Map of indicators summarised at district level.**
(TIFF)

## Author contributions

**Conceptualization:** William Jones-Warner, Joseph Opare, Richard Odame Phillips, Rachel Pullan, Hope Simpson.

**Data curation:** William Jones-Warner, Hope Simpson.

**Formal analysis:** William Jones-Warner, Hope Simpson.

**Funding acquisition:** Rachel Pullan.

**Investigation:** Richard Odame Phillips.

**Methodology:** William Jones-Warner, Yaw Ampem Amoako, Joseph Opare, Dorothy Yeboah-Manu, Richard Odame Phillips, Rachel Pullan, Hope Simpson.

**Project administration:** William Jones-Warner, Hope Simpson.

**Resources:** Yaw Ampem Amoako.

**Supervision:** Hope Simpson.

**Visualization:** William Jones-Warner.

**Writing – original draft:** William Jones-Warner.

**Writing – review & editing:** Yaw Ampem Amoako, Joseph Opare, Nana Konama Kotey, Dorothy Yeboah-Manu, Richard Odame Phillips, Rachel Pullan, Hope Simpson.

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
