## [Decision Letter · Decision Letter 0]

14 Jul 2025

PGPH-D-25-00462

Evidence Based Targeting of Districts for Active Surveillance of Skin-Related Neglected tropical diseases in Ghana

Dear Dr. Jones-Warner,

Thank you for submitting your manuscript to PLOS Global Public Health. After careful consideration, we feel that it has merit but does not fully meet PLOS Global Public Health’s publication criteria as it currently stands. Therefore, we invite you to submit a revised version of the manuscript that addresses the points raised during the review process.

Please note that we have only been able to secure a single reviewer to assess your manuscript. We are issuing a decision on your manuscript at this point to prevent further delays in the evaluation of your manuscript. Please be aware that the editor who handles your revised manuscript might find it necessary to invite additional reviewers to assess this work once the revised manuscript is submitted. However, we will aim to proceed on the basis of this single review if possible.

We look forward to receiving your revised manuscript.

Kind regards,

Jianhong Zhou

Staff Editor

Journal Requirements:

Additional Editor Comments (if provided):

Reviewers' comments:

Reviewer's Responses to Questions

**Comments to the Author**

1. Does this manuscript meet PLOS Global Public Health’s publication criteria ? Is the manuscript technically sound, and do the data support the conclusions? The manuscript must describe methodologically and ethically rigorous research with conclusions that are appropriately drawn based on the data presented.

Reviewer #1: Yes

2. Has the statistical analysis been performed appropriately and rigorously?

Reviewer #1: N/A

3. Have the authors made all data underlying the findings in their manuscript fully available (please refer to the Data Availability Statement at the start of the manuscript PDF file)?

Reviewer #1: Yes

4. Is the manuscript presented in an intelligible fashion and written in standard English?

Reviewer #1: Yes

Reviewer #1: As a clinician I am not familiar with many of the concepts in this manuscript and so found it rather hard going to read and review. If your target audience is aiming at professionals familiar with working with geographic data then no modification is required, but if you wish to engage the wider Skin NTD community to understand this proposed method and concept it might be worth revising the manuscript with the less geographic data aware individuals in mind.

For example:

What is a rapid literature review? I found this reference to explain it to me. Smela B, Toumi M, Świerk K, Francois C, Biernikiewicz M, Clay E, Boyer L. Rapid literature review: definition and methodology. J Mark Access Health Policy. 2023 Jul 28;11(1):2241234. doi: 10.1080/20016689.2023.2241234. PMID: 37533549; PMCID: PMC10392303.

You mention the presentation of data in a “raster” format. I had no idea what this was.

Malaria Atlas Project. Nature Africa Housing 2015 - 654 https://data.malariaatlas.org/maps?layers=Explorer:2019_Nature_Africa_Housing_201 655 5,Malaria:202206_Global_Pf_Parasite_Rate. 2023. this opens to a map and it is not useful to me as I did not really understand how to engage with it.

Demographic and Health Surveys Program. Demographic and Health Surveys Program 657 2017 Special Survey - 658 https://dhsprogram.com/data/dataset/Ghana_Special_2017.cfm?flag=0. 2023.

You mention the use of Demographic and Health Surveys (DHS) Program (reference 40) – as this is a proposed method for all countries, is the data collected uniform for each country? A quick look at the database, with which I am not familiar, indicated a variety of metrics for each country ie not all the same.

Page 20 visualisation ; you noted that you use “Rmarkdown”: should this be referenced?

Discussion

As I have absolutely no knowledge of R programming and/or GIS skills I find myself having to take the statements and assumptions reported at face value rather than being guided through them.

I hope these comments from a simple global health dermatologist clinician are helpful

**Do you want your identity to be public for this peer review?** For information about this choice, including consent withdrawal, please see our Privacy Policy .

Reviewer #1: No

---

## [Decision Letter · Decision Letter 1]

5 Jan 2026

PGPH-D-25-00462R1

Evidence Based Targeting of Districts for Active Surveillance of Skin-Related Neglected tropical diseases in Ghana

Dear Dr. Jones-Warner,

Thank you for submitting your manuscript to PLOS Global Public Health. After careful consideration, we feel that it has merit but does not fully meet PLOS Global Public Health’s publication criteria as it currently stands. Therefore, we invite you to submit a revised version of the manuscript that addresses the points raised during the review process.

Your article has been assessed by a statistical expert, who raises a number of concerns with the analyses conducted and the validation of your study. Could you please carefully revise the manuscript to address all comments raised?

We look forward to receiving your revised manuscript.

Kind regards,

Jen Edwards

Staff Editor

Journal Requirements:

Additional Editor Comments (if provided):

Reviewers' comments:

Reviewer's Responses to Questions

**Comments to the Author**

Reviewer #1: All comments have been addressed

Reviewer #2: (No Response)

Reviewer #3: All comments have been addressed

publication criteria ? Is the manuscript technically sound, and do the data support the conclusions? The manuscript must describe methodologically and ethically rigorous research with conclusions that are appropriately drawn based on the data presented.

Reviewer #1: Yes

Reviewer #2: Yes

Reviewer #3: Yes

3. Has the statistical analysis been performed appropriately and rigorously?

Reviewer #1: N/A

Reviewer #2: I don't know

Reviewer #3: Yes

4. Have the authors made all data underlying the findings in their manuscript fully available (please refer to the Data Availability Statement at the start of the manuscript PDF file)?

Reviewer #1: Yes

Reviewer #2: Yes

Reviewer #3: Yes

5. Is the manuscript presented in an intelligible fashion and written in standard English?

Reviewer #1: Yes

Reviewer #2: Yes

Reviewer #3: Yes

Reviewer #1: (No Response)

Reviewer #2: Very important topic and well-structured paper.

Reviewer #3: I write to submit my review on the manuscript titled “Evidence-Based Targeting of Districts for Active Surveillance of Skin-Related Neglected Tropical Diseases in Ghana.”

The authors analyzed secondary data using an evidence consensus approach to summarize and synthesize a variety of spatial datasets.

Here are my comments

1. Who were the local experts whose opinions were integrated into the overall study design? Kindly provide details of selection criteria and their experiences in BU and LF research, case management and surveillance, diagnosis and treatment etc

2. Kindly provide the predicted suitability range for BU. LF prevalence estimates will range from 0 to 100%. Was it the same for BU? Kindly state it. What is the definition of suitability range?

3. Line 325-327 is a bit confuse. “These two metrics were used due to the shared endemicity in Ghana, using both could indicate the co-endemicity of other skin NTDs and the ready availability of the datasets”. Kindly revise the statement. I understand that the availability of data is one of the reasons only two metrics were considered in selecting the districts but how it is currently phrased is a bit confusing. This should be clearly stated. You may revise as follows: These two metrics were used due to the ready availability of the datasets and shared endemicity in Ghana; using both could indicate the co-endemicity of other skin NTDs.

4. “We calculated mean values for each of the 260 districts(50)”. Mean values of what?

5. What informed the threshold of excluding districts in the bottom quintile (with predicted prevalence of 5% or less)? What would have been the implications on the statistical analyses if those districts with a predicted prevalence of 5% or less had been included in the analysis? These details and explanations must be provided in the manuscript

6. Line 343-344: “We considered both of these indicators together to identify where BU and LF overlapped.” How did you combine the two indicators to identify overlap? Because line 345-348 only talks about how the two distributions were converted to quintiles separately

7. Kindly include the implications for conducting single mean imputation using estimates from neighbouring districts instead of a more robust multiple imputation by chain equations

8. The DHS data sets are not powered at the district level and therefore wealth estimates will not be representative of the targeted districts. So, assessing vulnerability using DHS wealth quintiles at the district level is problematic as the sample is not representative of the targeted districts. DHS is stratified at the regional and rural, urban, and not powered at the district level, as the focus of the DHS is to generate national, regional and rural-urban stratification estimates. These limitations must be clearly highlighted and its implications on the current findings stated.

9. What informed this definition? “We defined co-endemic districts as those with any area predicted suitable for BU, and with a mean predicted LF antigen prevalence of at least 5%.” Is it based on the literature or is a standard definition? Reference. It was chosen for the purposes of this study, then kindly provide justification.

10. “Domain scores were then adjusted so that all domains carried equal weight, regardless of the number of component indicators” How did you adjust the domain score? Statistical method used for the adjustment?

11. Major concern: There was no internal validation of the methodology to assess robustness in real application. Authors indicated that “Six districts, including Shama, Awutu Senya East, and Ekumfi, scored high for both BU and LF, making them priority areas for active BU detection”. What is the observed number of BU and LF cases in the six priority districts compared to the over 100 districts in Ghana? If the methodology can identify high-priority areas, then internal validity assessment of the approach should demonstrate that the true observed cases of BU and LF in the six districts exceed the true observed cases of BU and LF from other districts in Ghana. This detailed internal validity assessment was completely missing from the manuscript.

**Do you want your identity to be public for this peer review?** For information about this choice, including consent withdrawal, please see our Privacy Policy .

Reviewer #1: **Yes:** Dr Lucinda Claire FullerDr Lucinda Claire Fuller

Reviewer #2: No

Reviewer #3: No

---

## [Decision Letter · Decision Letter 2]

13 Feb 2026

Evidence Based Targeting of Districts for Active Surveillance of Skin-Related Neglected tropical diseases in Ghana

PGPH-D-25-00462R2

Dear Mr Jones-Warner,

We are pleased to inform you that your manuscript 'Evidence Based Targeting of Districts for Active Surveillance of Skin-Related Neglected tropical diseases in Ghana' has been provisionally accepted for publication in PLOS Global Public Health.

Best regards,

Julia Robinson

Executive Editor

Reviewer Comments (if any, and for reference):

Reviewer's Responses to Questions

**Comments to the Author**

Reviewer #1: All comments have been addressed

Reviewer #3: All comments have been addressed

publication criteria ? Is the manuscript technically sound, and do the data support the conclusions? The manuscript must describe methodologically and ethically rigorous research with conclusions that are appropriately drawn based on the data presented.

Reviewer #1: Yes

Reviewer #3: Yes

3. Has the statistical analysis been performed appropriately and rigorously?

Reviewer #1: Yes

Reviewer #3: Yes

4. Have the authors made all data underlying the findings in their manuscript fully available (please refer to the Data Availability Statement at the start of the manuscript PDF file)?

Reviewer #1: Yes

Reviewer #3: Yes

5. Is the manuscript presented in an intelligible fashion and written in standard English?

Reviewer #1: (No Response)

Reviewer #3: Yes

Reviewer #1: I note that the first and last author are not from Ghana. It would be more appropriate in a prestigious global health journal to attribute a first and last author from the host LMIC nation, or at least joint first and last author. Please consider this.

Reviewer #3: The authors have comprehensively addressed all the comments and concerns raised in my previous review of the manuscript

**Do you want your identity to be public for this peer review?** For information about this choice, including consent withdrawal, please see our Privacy Policy .

Reviewer #1: **Yes:** Dr Lucinda Claire FullerDr Lucinda Claire Fuller

Reviewer #3: No
